# The Risk of Sarcopenia among Adults with Normal-Weight Obesity in a Nutritional Management Setting

**DOI:** 10.3390/nu14245295

**Published:** 2022-12-13

**Authors:** Antonino De Lorenzo, Massimo Pellegrini, Paola Gualtieri, Leila Itani, Marwan El Ghoch, Laura Di Renzo

**Affiliations:** 1Section of Clinical Nutrition and Nutrigenomic, Department of Biomedicine and Prevention, University of Tor Vergata, Via Montpellier 1, 00133 Rome, Italy; 2Department of Biomedical, Metabolic and Neural Sciences, University of Modena and Reggio Emilia, 41125 Modena, Italy; 3Department of Nutrition and Dietetics, Faculty of Health Sciences, Beirut Arab University, Riad El Solh, Beirut 11072809, Lebanon

**Keywords:** body composition, BMI, body fat, DXA, normal body weight, obesity, sarcopenia, sarcopenic obesity

## Abstract

Normal-weight obesity (NWO) is a phenotype characterized by excessive body fat (BF) despite normal body weight. We aimed to assess the association between NWO and the risk of sarcopenia. Two groups of patients with a normal body mass index [BMI (20–24.9 kg/m^2^)] were selected from a large cohort of participants. Body composition was measured using dual-energy X-ray absorptiometry (DXA), and 748 participants were categorized as NWO or normal-weight without obesity (NWNO) and were classed according to whether or not they were at risk of sarcopenia. The “NWO group” included 374 participants (cases), compared to 374 participants (controls) in the “NWNO group”, all of a similar BMI, age and gender. The participants in the “NWO group” displayed a higher prevalence of the risk of sarcopenia than the control group across both genders (0.6% vs. 14.1% in males; 1.4% vs. 36.5% in females). Regression analysis showed that being in the NWO category increased the risk of sarcopenia 22-fold in males (RR = 22.27; 95%CI: 3.35–147.98) and 25-fold in females (RR = 25.22; 95%CI: 8.12–78.36), compared to those in the NWNO category. In a “real-world” nutritional setting, the assessment of body composition to identify NWO syndrome is vital since it is also associated with a higher risk of sarcopenia.

## 1. Introduction

The World Health Organization (WHO) released a report on obesity prevalence in Europe in May 2022, stating that nearly 60% of citizens in this area are either affected by overweight or obesity [1]. Obesity is considered a major risk factor for several medical [2,3] and psychosocial morbidities [4,5,6,7], as well as increased rates of mortality [8]. Therefore, from a preventive point of view, its accurate identification in the early stages is vital [9]. 

Obesity is best defined as an excessive fat deposition in the adipose tissue [10,11]; consequently, its identification, based on BF quantification, is the gold standard and appears to be the most accurate [12]. Despite this, the adult obesity classification according to the WHO relies on a BMI cut-off point. More specifically, a BMI ≥ 30 kg/m^2^ in a Caucasian population is indicative of obesity, and this cut-off point is valid across all age groups and genders [10]. However, it should be kept in mind that BMI classification has several major limitations [13]. Firstly, it is unable to differentiate between BF and muscle mass, but more importantly, this cut-off point (i.e., BMI ≥ 30 kg/m^2^) becomes useless, especially in certain obesity phenotypes, such as NWO [14], whereby an individual falls within normal body weight status but has a high percentage of BF, characteristic of obesity.

The NWO syndrome has received particular attention from researchers over the last decade and has been widely studied, particularly in relation to females across all age groups [15,16]. The prevalence of NWO among the general population is nearly 10% [15,17], but this increases to more than 30% among patients with inflammatory-related diseases [18]. Individuals with NWO syndrome were found to be associated with higher cardiometabolic risk factors and increased odds of having dyslipidemia, hypertension, type 2 diabetes and metabolic syndrome, with respect to their relative NWNO [19]. Despite this fact, certain research areas among this population have not been studied and require further investigation. For instance, another topic which has recently received particular attention among the clinical population within a nutritional setting, especially in relation to patients with obesity, is sarcopenia [20], represented by an increased accumulation of BF and decreased muscle mass and strength. Sarcopenia is known to be a prevalent condition in patients classified as having obesity [21], according to the WHO BMI cut-off point (i.e., ≥30 kg/m^2^), particularly among females [20]. Moreover, sarcopenia is also known to be more closely associated with several medical [21] and psychosocial [22] comorbidities, when compared to those patients with obesity but not sarcopenia. However, it is still unclear whether sarcopenia affects individuals with NWO, and if this is the case, to what extent. Therefore, the purpose of this study was to assess the risk of sarcopenia among patients with NWO within a clinical nutritional setting.

## 2. Materials and Methods

### 2.1. Participants and Design of the Study 

The study is a single-center observational study. Subjects were pooled from a cohort, previously admitted to the Nutritional Unit at the Department of Biomedicine and Prevention, University of Rome “Tor Vergata”, Italy during the period June 2018–May 2022, as described elsewhere [23]. The inclusion criteria were (i) an age ≥ 20 years and (ii) a normal body weight status, according to the WHO BMI cut-off points, ≥18.5 kg/m^2^ and ≤24.9 kg/m^2^. Each patient had to complete a body composition test by means of DXA and was classified as being affected by obesity, based on age- and gender-specific BF percentage cut-off points as follows [24,25,26]: 20–39 years: BF% ≥ 39% for females and BF% ≥ 25% for males.40–59 years: BF% ≥ 40% for females and BF% ≥ 28% for males.60–79 years: BF% ≥ 42% for females and BF% ≥ 30% for males.

Patients were excluded if they were aged <20 years, were underweight, overweight or were suffering from obesity, based on the WHO BMI cut-off points (i.e., <20 kg/m^2^; >25 kg/m^2^), were pregnant or lactating, were taking medication that affects body weight or composition, presented with medical comorbidities associated with weight loss (e.g., cancers) or were suffering from severe psychiatric disorders. Accordingly, a total of 374 individuals representing both genders were included in the NWO group. The control group with a 1:1 ratio included 374 participants from the same cohort, with a similar BMI and the same gender distribution, but were classified as having a normal weight, based on age- and gender-specific BF cut-off points [24]. The study was approved by the Ethics Committee of the Calabria Region Center Area Section (No. 146 17/05/2018) and was conducted in accordance with the Declaration of Helsinki. All patients gave informed written consent.

### 2.2. Body Weight and Height

The height and weight of patients were measured using a stadiometer and an electronic weighing scale (SECA 2730-ASTRA, Hamburg, Germany). The BMI of each patient was then determined according to the standard formula of body weight (kg) divided by height (m) squared [27].

### 2.3. Body Composition

As described previously [23], a DXA scanner (DXA, GE Medical Systems) was used to determine body composition, which assessed both total and segmental compartments in terms of fat and lean mass, according to a standardized procedure, as described elsewhere [28]. The risk of sarcopenia was defined as appendicular skeletal muscle mass (ASM) (kg), adjusted for body weight (kg) (ASM/weight) [29,30,31]. The cut-offs obtained from the reference group were <0.2827 for males and 0.2347 for females [32].

### 2.4. Statistical Analysis

Descriptive statistics are reported as mean and standard deviations for continuous variables and frequencies, as well as proportions for categorical variables, while stratifying by gender. Means, frequencies and proportions were compared using a student’s *t* test and a Chi-squared test, respectively. To calculate the adjusted relative risk of sarcopenia among those with NWO or NWNO stratified by gender, Poisson regression analysis was used with a robust estimator, adjusting for age and BMI. All tests were considered significant at a *p* value < 0.05.

## 3. Results

A total of 748 participants were included in the analysis, comprising 326 males (43.6%) and 422 females (56.4%). The mean overall age was 36.78 ± 12.85 years, with a similar age distribution between the categories of normal weight and across genders. Of the male participants in both the NWO and NWNO categories, most were between 20 and 39 years (63.8% vs. 64.4%), respectively, almost one third were between 40 and 59 years (27.6% vs. 26.4%) and less than 10% were between 60 and 79 years (8.6% vs. 9.2%) (X^2^ = 0.085; *p* = 0.959). Females also had a similar age distribution in the NWO and NWNO categories, with most of them aged between 20 and 39 years (58.8% vs. 57.8%), respectively; almost one third were between 40 and 59 years (37.4% vs. 38.9%) and less than 5% were between 60 and 70 years (3.8% vs. 3.3%) (X^2^ = 0.139; *p* = 0.933). As for anthropometrics, those with NWO or NWNO had the same mean BMI among males (23.48 ± 1.10 vs. 23.49 ± 1.13 kg/m^2^) and females (23.28 ± 1.08 vs. 23.38 ± 1.09 kg/m^2^), respectively. In terms of body composition, those with NWO had a significantly higher mean in terms of total BF, compared to those with NWNO among both the male (21.67 ± 3.12 vs. 11.85 ± 2.78 kg) and female categories (26.07 ± 2.88 vs. 18.26 ± 2.62 kg). Similarly, the mean BF percentage was significantly higher in the NWO category, compared to the NWNO category among both males (29.68 ± 3.25% vs. 16.42 ± 3.84%) and females (41.90 ± 2.16% vs. 29.89 ± 3.09%), respectively. The ASM was also significantly lower in the NWO category, compared to the NWNO category among males (22.22 ± 2.90 vs. 26.99 ± 3.47) and females (14.86 ± 1.73 vs. 17.90 ± 2.42), respectively. Similarly, the ASM as a percentage of total body weight was significantly lower in the NWO category, compared to the NWNO category in the case of both males (30.40 ± 2.58% vs. 37.18 ± 2.71%) and females (23.89 ± 1.62% vs. 29.27 ± 2.19%), respectively (Table 1).

The risk of sarcopenia among males and females in both the NWO and NWNO categories is presented in Table 2. Among the males, the total number of participants with NWO and with a higher risk of sarcopenia was 23 out of 326 or 14.1%. Among the females, the total number of participants with NWO and at risk of sarcopenia was 77 out of 211 or 36.5%. 

Poisson regression analysis showed that being in the NWO category increased the risk of sarcopenia 22-fold in males (RR = 22.27; 95%CI: 3.35–147.98) and 25-fold in females (RR = 25.22; 95%CI: 8.12–78.36), compared to those in the NWNO category, while keeping age and BMI constant (Table 3). 

## 4. Discussion

The current study aimed to provide preliminary data relating to the prevalence of the risk of sarcopenia in patients with NWO, by comparison with NWNO in a nutritional clinical setting.

### 4.1. Findings and Concordance with Previous Studies

Our main finding was that the risk of sarcopenia in individuals with NWO in a clinical setting is high across both genders (14.1% in males and 36.5% in females), whereas the presence of the latter (i.e., NWO) is associated with a risk of sarcopenia among individuals with apparent normal weight. Even if we are not in a position to describe the mechanism behind this association, we may, however, speculate on this; in fact, we believe that there are common denominators between NWO and sarcopenia [33], one of which can be inflammation [34]. Certain studies have reported higher levels of proinflammatory markers (i.e., TNFα, IL-1α, IL-1β, IL-6, and IL-8, etc.) in the NWO category, compared with the NWNO category (i.e., with a normal BF percentage) [15,35]. Similarly, the association between high inflammatory markers and sarcopenia has been widely confirmed, whereby several inflammatory cytokines have been shown to prompt muscle wasting by stimulating protein catabolism and suppressing muscle synthesis [36]. Therefore, it would appear that there is a cross reference between adipose tissue and skeletal muscle through inflammation, as a main mechanism of the pathogenesis of the correlation between NWO and sarcopenia [37].

### 4.2. Potential Clinical Implications and New Directions 

Our findings have a certain implication, primarily identifying the hidden obesity in apparently normal weight patients in clinical nutrition settings. More awareness should be raised among healthcare professionals regarding the existence of this condition (i.e., NWO) in clinical settings, as recognizing its association with a higher risk of sarcopenia is vital. In addition, certain new directions for future research are required. Firstly, research needs to confirm our findings in larger studies. Secondly, the potential negative impact of the combination of sarcopenia and NWO on health outcomes (medical and psychosocial) with respect to NWNO individuals requires further investigation. Finally, a better understanding of the precise mechanism behind the high risk of sarcopenia among this specific population (i.e., NWO) is vital (i.e., proinflammatory state) from both a preventive and curative perspective. 

### 4.3. Study Strengths and Limitations 

Our study has certain strengths. To the best of our knowledge, it is the first to investigate the risk of sarcopenia in a clinical population that falls within the normal range, according to the WHO BMI cut-off points (i.e., 20–24.9 kg/m^2^) but is affected by obesity, based on age- and gender-specific cut-off points in relation to BF percentages. The study includes adults of both genders in a ‘real-world’ clinical setting in Italy. Secondly, body composition was measured using DXA, which is considered to represent a precise method [38] and is regarded as the gold standard by several populations [39]. However, our study also has certain limitations. Firstly, we did not perform any functional test to assess muscle strength–which is considered as another necessary component besides muscle mass–in order to confirm the presence/absence of sarcopenia. In fact, our study discussed the risk of sarcopenia (i.e., as a reduction of muscle mass) rather than “sarcopenia” as a confirmed diagnosis [40]. Secondly, we used a small sample size, and data were collected in a single unit, requiring external validation across other populations since our results derive from a single unit [41]. Finally, we did not perform an objective assessment of dietary intake and physical activity levels, factors known to affect body composition.

## 5. Conclusions

NWO syndrome is a hidden condition prevalent in clinical settings and known to be associated with high cardiovascular risk, insulin resistance, oxidative stress and a pro-inflammatory state [14,19]. In our study, we demonstrated that this condition is also associated with the risk of sarcopenia in a normal weight population that may appear “healthy” in the first instance. For this reason, our findings emphasize that body composition measurements should be routinely assessed in nutritional settings, even among patients with a normal weight status, so as to identify whether they have the NWO phenotype, as well as the presence of sarcopenia. 

## Figures and Tables

**Table 1 nutrients-14-05295-t001:** Demographic and anthropometric characteristics of the study participants ^¥^ (*n* = 748).

	Total *n* = 748	Males *n* = 326			Females *n* = 422		
		NWNO *n* = 163	NWO *n* = 163	Significance	NWNO *n* = 211	NWO *n* = 211	Significance
Age	36.78(12.85)	36.18(13.51)	36.82(13.34)	0.671	36.68(12.45)	37.29(12.38)	0.613
				X^2^ = 0.085; *p* = 0.959			X^2^ = 0.139; *p* = 0.933
20–39	455(60.8)	104(63.8)	105(64.4)		124(58.8)	122(57.8)	
40–59	249(33.3)	45(27.6)	43(26.4)		79(37.4)	82(38.9)	
60–79	44(5.9)	14(8.6)	15(9.2)		8(3.8)	7(3.3)	
BMI (kg/m^2^)	23.40(1.10)	23.48(1.10)	23.49(1.13)	0.94	23.28(1.08)	23.38(1.09)	0.367
BF (kg)	19.81(5.86)	11.85(2.78)	21.67(3.12)	<0.0001	18.26(2.62)	26.07(2.88)	<0.0001
BF%	30.30(9.47)	16.42 (3.84)	29.68 (3.25)	<0.0001	29.89(3.09)	41.90(2.16)	<0.0001
				X^2^ = 326; *p* < 0.0001			X^2^ = 422; *p* < 0.0001
NWNO	374(50.0)	163(100.0)	0(0.0)		211(100.0)	0(0.0)	
NWO	374(50.0)	0(0.0)	163(100.0)		0(0.0)	211(0.0)	
ASM	19.97(5.23)	26.99(3.47)	22.22(2.90)	<0.0001	17.90(2.42)	14.86(1.73)	<0.0001
ASM%	29.72(5.20)	37.18(2.71)	30.40(2.58)	<0.0001	29.27(2.19)	23.89(1.62)	<0.0001

^¥^ Values are *n* (%) for categorical variables and Mean (SD) for continuous variables; NWNO = normal weight non obesity; NWO = normal weight obesity; BMI = body mass index; BF = body fat; ASM = appendicular skeletal mass.

**Table 2 nutrients-14-05295-t002:** The risk of sarcopenia in NWNO and NWO across gender ^¥^ (*n* = 743).

	Total *n* = 748	Males *n* = 326			Females *n* = 422		
		NWNO *n* = 163	NWO *n* = 163	Significance	NWNO *n* = 211	NWO *n* = 211	Significance
No risk of sarcopenia	644(86.1)	162 (99.4)	140 (85.9)	X^2^ = 21.77; *p* < 0.01	208(98.6)	134(63.5)	X^2^ = 84.46; *p* < 0.01
Risk of Sarcopenia	104(13.9)	1 (0.6)	23 (14.1)		3 (1.4)	77 (36.5)	

^¥^ NWNO = normal weight non obesity; NWO = normal weight obesity.

**Table 3 nutrients-14-05295-t003:** Adjusted relative risk for sarcopenia ^¥^ (*n* = 743).

	Males *n* = 326			Females *n* = 422		
	RR	95%CI	Significance	RR	95%CI	Significance
Age	1.07	1.05–1.09	<0.00001	1.02	1.01–1.04	0.001
BMI	0.77	0.56–1.05	0.102	1.05	0.89–1.23	0.573
BF%						
Normal	1			1		
With obesity	22.27	3.35–147.98	0.001	25.22	8.12–78.36	<0.0001

^¥^ Models are adjusted for age and BMI.

## Data Availability

The dataset in the present study is available upon request.

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
