# Peer review of "The Risk of Sarcopenia among Adults with Normal-Weight Obesity in a Nutritional Management Setting"

_nutrients, 2022, doi:10.3390/nu14245295_

Round 1
Reviewer 1 Report
This paper showed the prevalence of normal weight obesity in real-world setting using DXA. The information of normal weight obesity especially for sarcopenic obesity should be important, and this paper could give us useuful information about this issue. Although this research seems to be rather preliminary, but still worthwhile to sharing the information of the result of this paper.
1. Authors must describe the precise criteria to define as obesity by body fat percentage cut off in materials and method.
2. As this journal is "Nutrients", we would like to have nutritional information in the study. If they don't have enough information to show in this study, they should add it as a limitation of this study.
Author Response
1. Authors must describe the precise criteria to define as obesity by body fat percentage cut off in materials and method.
Response: Done as suggested (Page 2, paragraph 3).
2. As this journal is "Nutrients", we would like to have nutritional information in the study. If they don't have enough information to show in this study, they should add it as a limitation of this study.
Response: Done as suggested and added as limitation in in the Discussion section (Page 5, paragraph 4).
Reviewer 2 Report
The results of the paper are in agreement with other studies that show that those with increased body fat have less muscle or lean mass. However, we do not have strong evidence that obesity causes muscle loss. As people age, they lose muscle and gain body fat. The loss of muscle actually slows down metabolism resulting in increased body fat. Therefore, sarcopenia can also lead to obesity. In line 158 you state that normal weight obesity (NWO) greatly increases the risk of sarcopenia. You do not have enough evidence to state this so please remove this sentence.
Secondly, sarcopenia is not just loss of muscle, but is defined as loss of muscle and STRENGTH with age. In the paper there is no reference to measuring strength such as using a knee extension machine or weights to assess a participant's maximum strength etc. https://my.clevelandclinic.org/health/diseases/23167-sarcopenia
Please re-word your paper to correctly define sarcopenia and list as a limitation the salient fact that you did not measure strength or loss of strength.
Author Response
The results of the paper are in agreement with other studies that show that those with increased body fat have less muscle or lean mass. However, we do not have strong evidence that obesity causes muscle loss. As people age, they lose muscle and gain body fat. The loss of muscle actually slows down metabolism resulting in increased body fat. Therefore, sarcopenia can also lead to obesity. In line 158 you state that normal weight obesity (NWO) greatly increases the risk of sarcopenia. You do not have enough evidence to state this so please remove this sentence.
Response: We reworded the statement to appear as an association to exclude any cause-effect relationship between the two conditions (Page 6, paragraph 1).
Secondly, sarcopenia is not just loss of muscle, but is defined as loss of muscle and STRENGTH with age. In the paper there is no reference to measuring strength such as using a knee extension machine or weights to assess a participant's maximum strength etc. https://my.clevelandclinic.org/health/diseases/23167-sarcopenia. Please re-word your paper to correctly define sarcopenia and list as a limitation the salient fact that you did not measure strength or loss of strength.
Response: Done as suggested. We reworded our paper based on the reviewer’s valuable comment. In fact, our study discussed the risk of sarcopenia (i.e. Considering as so those with a reduced muscle mass) rather than “sarcopenia” as a confirmed diagnosis (i.e. reduced muscle mass and strength). As well as added as limitation in in the Discussion section (Page 5, paragraph 4).
Moderate English changes required
Response: Done as suggested.
Round 2
Reviewer 2 Report
Nice revisions.